# Nonlinear Statistical Features of the Seismicity in the Subduction Zone of Tehuantepec Isthmus, Southern México

**DOI:** 10.3390/e24040480

**Published:** 2022-03-30

**Authors:** Alejandro Ramírez-Rojas, Elsa Leticia Flores-Márquez

**Affiliations:** 1Departamento de Ciencias Básicas, Universidad Autónoma Metropolitana, Azcapotzalco, Mexico City 02200, Mexico; 2Instituto de Geofísica, Universidad Nacional Autónoma de México, Circuito Institutos s/n, C.U., Coyoacán 04510, Mexico

**Keywords:** seismicity, Isthmus of Tehuantepec, Gutenberg–Richter law, nowcasting, multifractal detrended fluctuation analysis, visibility graph

## Abstract

After the M8.2 main-shock occurred on 7 September 2017 at the Isthmus of Tehuantepec, Mexico, the spatial distribution of seismicity has showed a clear clusterization of earthquakes along the collision region of the Tehuantepec Transform/Ridge with the Middle America Trench off Chiapas. Furthermore, nowadays, the temporal rate of occurrence in the number of earthquakes has also showed a pronounced increase. On the basis of this behavior, we studied the sequence of magnitudes of the earthquakes which occurred within the Isthmus of Tehuantepec in southern Mexico from 2010 to 2020. Since big earthquakes are considered as a phase transition, after the M8.2 main-shock, one must expect changes in the Tehuantepec ridge dynamics, which can be observed considering that the b-value in the Gutenberg–Richter law, has also showed changes in time. The goal of this paper is to characterize the behavior of the seismic activity by using the Gutenberg–Richter law, multifractal detrended fluctuation analysis, visibility graph and nowcasting method. Those methods have showed important parameters in order to assess risk, the multifractality and connectivity. Our findings indicate, first that b-value shows a dependency on time, which is clearly described by our analyses based on nowcasting method, multifractality and visibility graph.

## 1. Introduction

The dynamical processes into the Earth’s interior drives the movement of the tectonic plates following a complex dynamic. The seismic activity is fed continuously with energy obtained from the tectonic plate dynamics. The earthquakes occur as an energy dissipation process in the Earth’s crust, just a phase transition, to which the tectonic energy is continuously inserted. According to Bak et al. (1988), the tectonic plates self-organize toward critical states, allowing temporal and spatial fractals structures to emerge naturally, and the power-laws are the natural expression to describe such critical states [1,2]. Additionally, the crust therefore attains self-organized critically (SOC) states, analogous with the states of the sandpile model proposed by Bak et al. (1988) [2]. From a seismic point of view, the system evolves through a sequence of states which are usually referred to as seismic processes, in fact, are energy fluctuation processes, where energy is released in temporal periods interspersed with low activity and with events able to release large energy [3]; therefore, the seismicity can be described as a self-affine process whose dynamical variables are the magnitude (or energy) and the inter-event times of earthquakes. Many earthquake sequences in the world have been studied by applying different methods, for example for inter-event time series [4,5,6,7,8,9,10] and sequences of magnitudes (or energy fluctuation time series) [11,12,13,14,15,16,17] have been analyzed by using fractal/multifractal methods among others. The natural sources of earthquakes are faults, subduction zones and volcanoes, being the subduction zones of great seismological interest, because the greatest number of earthquakes occurs in those subduction zones; in fact, all of those zones in the world are subject to continuous monitoring. The largest number of earthquakes produced in the subduction zones are, mainly, interplate earthquakes, whose main mechanism is stick-slip, but earthquakes also occur, albeit less frequently, in the intraplate areas. The M8.2 earthquake that occurred in the Isthmus of Tehuantepec, on 7 September 2017, was considered as an unusual event [14] because the epicenter was in the intraplate. The stick-slip is the principal mechanism involved in the subduction dynamics, where the plates’ age, relative velocity between plates, dip angle and their own rough structure of the local zone could be determinant in the seismic activity [18,19]. The subduction mechanism could be explained as a complex dynamical system, and then attending to what the SOC theory [2] explains, the possibility of predicting earthquakes is scarce (see also [20]). Nevertheless, there are numerous published works dealing on the possibility of statistically identifying precursory signals associated with earthquakes; these signals are associated with the state of stress of the area where a rupture will occur, and they can be linked to changes in the electric and magnetic fields of the subsoil [21,22,23,24,25]; analyses in natural-time of data series in particular have been very successful in such characterizations [24]. The same type of analysis has been used for the characterization of data point processes such as earthquakes, which has allowed the use of the nowcasting method to assess the risk level of an earthquake that exceeds a given magnitude.

The Mexican Pacific coast is a region with important seismic activity that occurs mainly in a subduction zone. The subduction zone, located in southern Mexico, is approximated by a sub-horizontal slab bound at the edges by the steep subduction geometry of the Cocos plate, beneath the Caribbean plate to the east, and of the Rivera plate, beneath North America to the west [26]. Part of this subduction zone is the Isthmus of Tehuantepec, located near the triple complex junction formed by the North American, Cocos and Caribbean plates [27]. Many authors have studied this zone from different points of view, analyzing different parameters of subduction regimes. In a recent study, [28] presented the first seismic velocity model of the Tehuantepec subduction zone using the enhanced seismic tomography method, which allowed researchers to reconstruct the geometry of the slab and surroundings of this complex sector of Mexico.

The monitored seismic activity is showed in Figure 1 and Figure 2. Figure 1 shows the cumulative temporal rate by year, from which we can see that from 2010 to 2015, its behavior is defined by low and regular increases, which is approximately homogenous for 6 years. In addition, we can see that the spatial distribution of earthquakes in Figure 2, before the M8.2 quake, does not show alignment groupings. From 2015 until 2017, when the M8.2 occurs, a notorious change in the cumulative temporal rate is observed, but both periods also behave as a linear rate.

After the M8.2 mainshock, the epicenters were distributed as a well-defined clusterization following the collision trajectory of the Tehuantepec Ridge with the Middle America Trench off Chiapas, as can be seen in Figure 2. In addition, the accumulated annual rate of seismicity was increased very fast (see Figure 1). The time span of such a fast tendency was six months, from September 2017 to March 2018. From April 2018, the rate of production of earthquakes by month becomes constant again, but with a considerable increase in the number of events compared to the period before the M8.2 earthquake.

Paying attention to this unusual behavior in the zone [29,30,31], in this paper we analyze the magnitude sequences of the monitored seismicity that occurred within the Isthmus of Tehuantepec, which geographically includes the Tehuantepec Gulf and the Oaxaca and Chiapas States. The analyzed period is from 1 January 2010 to 31 December 2020, which includes the last two large earthquakes, the M8.2 on 7 September 2017, and a M7.4 occurring on 23 June 2020, the latter striking within the Isthmus of Tehuantepec close to the Oaxaca coast. The behavior of the accumulated rate in the analyzed ten-year period suggests changes in the dynamic evolution in the region. In order to differentiate the three stages in the seismic activity along the studied period, we have divided the entire catalog into three sub-catalogs: the first from 1 January 2010 to 7 September 2017, the second from 8 September 2017 to 31 March 2018, and the third from 1 April 2018 to 31 December 2020; the first and the third are magnitude sequences of regular seismicity, whilst the second corresponds to the aftershock activity. For the sake of brevity, each sub-catalogue will be referred to as part 1, part 2 and part 3, respectively. Because aftershocks are usually triggered by the static stress change associated with the mainshock, as well as some other post-seismic relaxation processes such as afterslip [32], the nowcasting method was only applied to part 1 and part 3, whilst part 2 was fitted with the Utsu–Omori law for the aftershocks. In addition, we studied the following issues: the b-value temporal behavior of the Gutenberg–Richter law; the concept of how self-similarity of the seismic phenomenon sustains the fractal methods, which are often used to identify correlation properties [33], and their estimation is based on multifractal detrended fluctuation analysis (MFDFA) [34]; the connectivity, which was investigated by using the visibility graph (VG) [35]. These methods are explained in the methodology section and were applied to part 1, part 2 and part 3.

Our findings indicate that the b-value in the Gutenberg–Richter law, as well as the completeness magnitudes, changed for each sub-catalogue when yearly windows were considered. Even though the data series are from the same region, the nowcasting analysis suggests the characteristic hazard level is different in each part. On the other side, the multifractality is applied to study the variability on a wide range of temporal or spatial scales, where the generalized Hurst exponent, Hq, measures such variability and also the persistence. For the three sub-catalogues, it is observed that antipersistence and the multifractality in part 2 is less than that of parts 1 and 3; in addition, the width of the singularities distribution, calculated yearly from the multifractal detrended fluctuation analysis (MFDFA), is increasing, except for the year after the mainshock. 

## 2. Tectonic Setting

The Isthmus of Tehuantepec zone shows clear tectonic disruption evidence that has been studied by various authors. Here, the Cocos plate presents a major linear feature that lies almost perpendicular to the Middle American trench; in [16,36] the authors calculated ocean floor ages and spreading rates, and their morphostructural analysis of the ridge and the surrounding ocean floor were used to infer the tectonic evolution and pattern of the Tehuantepec Ridge and associated structures. Their investigation about spreading rate changes led them to propose the existence of a microplate. This microplate would have been bound by the Tehuantepec Ridge and by a pseudo-transform fault. Reference [37] studied the genesis of the Chiapas foldbelt, which has been linked to the inferred eastward relative movement of the Chortis block (mainly Honduras) from a position off southwestern Mexico during the last 45 Ma. Keppie and Moran-Zenteno (2005) [38] proposed that the fold-and-thrust belt resulted from collision of the Tehuantepec Ridge with the Middle America Trench (see Figure 9 reported in [37]). The studies of the exposed Tehuantepec Transform/Ridge show that it varies from a transform fault, across which the age of the oceanic crust changes, and it produces a step (down to the east) to a ridge resulting from compression following a change in plate motion and a series of seamounts. However, the interpretation of these authors differs from that recently published by Calo (2021) [28] from this Transform/Ridge zone, where there are also present vertical and horizontal tears and a break of the slab at depth. Other authors have studied different aspects in tectonism of this zone, and the main evidence encountered is as follows: (1) transition of the subduction pattern [39,40,41], (2) different convergence rate of the subduction [41,42] (De Mets et al., 1994; Müller et al., 2016) and (3) the abrupt change in the depths of the earthquakes which occurred between both sides of the Tehuantepec Transform/Ridge [43] and references therein. This evidence has led them to propose models considering the existence of vertical (or trench-orthogonal) tears in this portion of the Cocos plate; however, they discard horizontal (or trench-parallel) ones or slab detachments. The Calo study (2021) [28] put forward the existence of horizontal tears and bending; Calo’s seismic velocity model of the Tehuantepec subduction zone, using an enhanced seismic tomography method, shows a 3D reconstruction of the geometry of the slab and surroundings of Tehuantepec Transform/Ridge. He concludes that the Tehuantepec Ridge starts to break the slab only at depths greater than 120–130 km, producing an evident vertical tear at depths greater than 140 km; moreover, he observed a horizontal tear for the first time in the oldest portion of the Cocos slab at depths of 150–160 km. However, his study is not conclusive regarding a vertical tear expected in the Oaxaca-Tehuantepec transition zone. It is clear that the study of the seismicity of the Isthmus of Tehuantepec can still provide much information on the dynamics of this subduction zone.

## 3. Materials and Methods

### 3.1. Data Set 

The data set analyzed was obtained from the seismic catalog of the National Seismic Service (SSN) of the Universidad Nacional Autónoma de México (UNAM) (www.ssn.unam.mx (accessed on 20 January 2021)) within the period from 1 January 2010 to 31 December 2020. According to Figure 1, it can be distinguished into three sub-catalogues: the first from 1 January 2010 to 7 September 2017, so-named part 1; the second from 8 September to 31 March 2018, so-named part 2; the third from 1 April 2018 to 30 July 2020, so-named part 3. As is observed in Figure 1, a clear difference is observed in the distribution of earthquakes after the M8.2 quake, when the cumulative temporal rate increases notoriously. Additionally, from April 2018, the cumulative distribution rate decreased.

The analysis was performed by considering all earthquakes whose epicenters were located between 92.5 and 96.5 degrees of longitude and between 14.7 and 16.5 degrees of latitude (see Figure 2); within this area, the M8.2 and M7.4 earthquakes occurred on 7 September 2017 and 23 June 2020, respectively. This region lies within the Isthmus of Tehuantepec region, from Oaxaca to Chiapas States. 

### 3.2. The Gutenberg–Richter Law

The Gutenberg–Richter (GR) law describes the relationship between the frequency and magnitude (*M*) of earthquakes in a specific region (Gutenberg and Richter, 1954) [44].
(1)$$log_{10}\;N=a-bM$$
where *a* and *b* values are constants to be determined and *N* is the number of EQs having a magnitude ≥ *M*.

GR law has been the preferred way to characterize the statistical behavior of earthquakes, and the b-value was widely observed in many works to be equal 1 [45]; it has even been argued for the universality of b = 1 [46,47]. However, changes in the b-value have been observed that are related to the spatial location of the analyzed area and to the time span observed. Notably, the pattern of b-value variation depends strongly on area of surveillance. On the other hand, both laboratory studies of rock deformation [48,49,50] and studies of seism aftershock sequences [51,52,53] have showed that changes in b-value are inversely related to changes in stress. Moreover, an important study of variations in b-value before several large earthquakes in north China, conducted by Ma (1978) [54], has found that for smaller areas around the earthquake epicenters, b-values vary with time from higher to lower values as the earthquake approaches, while for larger areas, peak values of b appear immediately before the earthquakes. According to Ma [54], the time duration of that peak values seems to be related to the magnitude of the impending event. In addition, Ma [54] found a spatial variation of b-values, with areas of lower b-value in the vicinity of the earthquake compared to the areas surrounding the epicenter of the earthquake. However, not all temporal decreases in b-value are followed by a significant earthquake [55,56]. Other studies [57] proposed that asperities may be characterized by anomalously low b-values of the frequency-magnitude distribution (FMD), in contrast to high b-values along creeping segments of faults [58]. While other authors such Zuñiga and Wyss (2001) [59], believe that the b-value depends on the magnitude scale used, and consider that the estimation of the local recurrence times for a given maximum magnitude on that scale is not affected by the slope b. They also depict that locally, along the subduction region of the Mexican Pacific coast, the b-values change in time ranging from 0.5 to 1.5. 

Some proposed earthquake precursors include increases in seismicity rate in a wide area around the expected mainshock [60,61,62,63,64] as well as decreases in rate within the expected source volume [60,65,66,67]. To define these patterns, a reliable record of seismicity as a function of space, time, and magnitude is needed.

Recently, studies conducted on spring-block models [68,69] provided information on the seismic dynamics; in particular they showed the linear relationship between the a-value and b-value parameters of the Gutenberg–Richter (GR) law, being *a*~4*b*. They also found the anticorrelation between the b-value of the GR law and the elastic ratio parameter of one computational model, for a determined region.

### 3.3. Utsu–Omori Law (UO)

The characterization of the aftershocks is given by the power law (Equation (1)) and was proposed by Omori (1894) [70] and modified (Equation (2)) by Utsu (1961) [71]. The aftershocks are understood as an effect of the readjustment to the new stress state in the source volume of the regions within the rupture zone or next to it [72,73] and usually are seisms triggered by the static stress change associated with the mainshock, as well as some other post-seismic relaxation processes such as afterslip, which occurs after main shocks. Earthquakes greater than M7 generally have thousands of small seisms. As an overall rule, aftershocks denote minor readjustments along the portion of a fault that slipped at the time of the mainshock according to Maeda (1999) [74]. In 1894, Omori established an empirical relation that is as follows:(2)$$n\left(t\right)=\frac{K}{\left(c+t\right)}$$
where *n* is the number of earthquakes by time interval, *K* > 0, *c* > 0, and *t* ≥ 0 are constants depending on analyzed earthquake sequences. The modified version, Utsu–Omori’s law, which is more commonly used today, was proposed by Utsu in 1961, this relationship (UO) establishes the frequency of aftershocks diminishes as they hyperbolically decay with the time afterwards the main shock. It is established as
(3)$$n\left(t\right)=\frac{K}{{\left(c+t\right)}^{p}}$$
where *p* is a constant [73] that adjusts the decay rate and their values are usually between 0.5 and 1.5, and its average value is only slightly above unity: *p* = 1.08. Equations (1) and (2) are empirical relationships that allow us to make an estimation of the probability of future aftershock occurrence; the probability of an aftershock is also going to decrease very quickly, a behavior that can be seen in the aftershock’s statistics.

### 3.4. Multifractal Analysis 

Complex time series are characterized by variability on a wide range of temporal or spatial scales that can be associated with intermittent fluctuations and long-range correlations characterized by different scaling behaviors [33,34], corresponding to different interwoven fractal subsets; thus, more than just one scaling exponent is required to be fully described. When a time series is characterized by only one scaling exponent, this indicates that when a single singularity dominates the time series, it is called monofractal, but if there are many dominant singularities, the time series is called multifractal [12]. The multifractality is described by means of the generalized Hurst exponent, *H*(*q*), and the singularity spectrum *f*(*α*) [75]. The set of fractal dimensions immerses into the time series is determined by the set of Hölder exponents *α*-values. The multifractal detrended fluctuation analysis (MFDFA), introduced by Kantelhardt et al. (2002) [34], is a method efficiently used when time series contains nonstationarities components, whose origin and scales are often unknown with the advantages that MFDFA requires a simple implementation. The procedure is well described, for instance, in [34,76]. The MFDFA’s steps are as follows: The profile $Y\left(i\right)\equiv \overset{i}{\underset{k=1}{\sum }}\left[x_{k}-\langle x\rangle \right],\;$ is determined by the integration, where 〈x〉 is mean value of all time series. Then, the profile $Y\left(i\right)$ is divided in nonoverlapping windows, $N_{s}=\left[N/s\right]$, of equal length ***s***. In order not to disregard this part of the series, the same procedure is repeated starting from the last until the first value of the series. Thereby, 2*N****_s_*** segments are obtained altogether. The local trend is calculated by a least-squares fit of the series, and for each segment *n* the variance can be determined, with *n =* 1, 2, …, *N_s_*. The fluctuation function is, then estimated by the formula
(4)$$F^{2}\left(s,\nu \right)=\frac{1}{s}\overset{s}{\underset{\nu =1}{\sum }}{\left\{\left(Y\left[\left(\nu -1\right)s+i\right]-y_{\nu ,i}\right)\right\}}^{2}$$

Here, $y_{\nu ,i}$ is the fitting polynomial in segment *n* of a certain degree *m*. Next, the *q*th-order fluctuation function for *q* ≠ 0 is obtained as:(5)$$F_{q}\left(s\right)={\left\{\frac{1}{2N_{s}}\sum_{\nu =1}^{2N_{s}}{\left\{F^{2}\left(s,\nu \right)\right\}}^{\frac{q}{2}}\right\}}^{\frac{1}{q}}$$
while for *q* = 0
(6)$$F_{q\to 0}\left(s\right)\equiv F_{0}\left(s\right)=exp\left\{\frac{1}{4Ns}\sum_{\nu =1}^{2Ns}\ln\left[F^{2}\left(s,\nu \right)\right]\right\}\sim s^{h\left(q=0\right)}.$$

The scaling behavior is determined from the fluctuation functions according to the power law: (7)$$F_{q}\left(s\right)\sim s^{h\left(q\right)}$$

The multifractal distribution or singularity spectrum obtained as $f\left(\alpha \right)=q\alpha -\tau \left(q\right)=q\left[\alpha -H\left(q\right)\right]+1\;$ where $\tau \left(q\right)=qH\left(q\right)-1$ and $\frac{d\tau }{dq}=\alpha$ are the Legendre transform. In this method, *α* is the Hölder exponent and *f*(*α*) indicates the dimension of the subset of the series that is characterized by *α*. The multifractal spectrum gives information about the relative dominance of various fractal exponents present in the series. In particular, the width of the spectrum indicates the range of the fractal exponents, so the larger the width, the more multifractal the series [13].

### 3.5. Visibility Graph Analysis

To assess the connectivity of a time series, the visibility graph (VG) approach was developed by Lacasa et al. (2008) [35]. The method maps a time series into a graph or network, with the advantage that the dynamical properties of the time series become topological properties of the graph or network, so that it is possible to uncover information of the time series just by analyzing the network’s topological properties; for instance, the degree distribution is related to the connectivity of the time series. The construction of the VG is performed with univariate time series of values that represent a scalar observable *y_i_*, recorded at times *t_i_*, that is, the series is considered as a sequence of pairs (*t_i_*, *y_i_*). In the VG approach, the *y_i_*-values are represented as nodes (or vertices) in the graph and separated by distances given by the *t_i_*. The graph or network must be well defined by a relation between nodes that they have to satisfy. Lacasa et al. (2008) [35] proposed the VG relation given when two nodes are mutually connected when a pair of nodes can see each other by a straight line, which means that such a segment is not broken by any other intermediate value of the series [61]. The relation between nodes that allows determine the connectivity is defined by the following rule:(8)$$y_{c}<y_{b}-\left(y_{a}-y_{b}\right)\frac{t_{b}-t_{c}}{t_{b}-t_{a}}$$
where the pairs (*t_a_, y_a_*), (*t_b_, y_b_*) and (*t_c_, y_c_*) represent three events occurred on times *t_a_* < *t_c_* < *t_b_*. 

Within the seismicity context, *y_i_* represents the magnitude of an earthquake occurred at time *t_i_* as it appears in any catalogue. The connectivity in graph theory means a path between every pair of vertices, and the degree *k* is the number of edges or paths incident to a vertex which is a measure of the connectivity, so that the connectivity depends on the dynamical features of the time series associated with the studied system. In addition, the degree distribution represents a property associated with the fractality features; for instance, for time series obtained from fractional Brownian motions, the degree distribution *P*(*k*) keeps a power-law relationship $P\left(k\right)\propto k^{-\gamma }$ where the exponent γ is related to the Hurst exponent [35,77].

### 3.6. Nowcasting Method

This method allows us to estimate the current hazard level in an seismically active region. The so-called nowcasting method was introduced by Rundle et al. [78]; it describes the present state of a system by counting the number of small earthquakes occurring within the elapsed time between two large earthquakes in a defined region [79]. This counting is linked to the earthquake cycle, because the absolute stress and strain since a last major earthquake cannot be determined from direct observations at all locations of interest [78]. This methodology analyzes seismic catalogues by using the natural time introduced by Varotsos et al. [80,81], in the frame of which an order parameter has been defined, the study of the variability of which [82,83] has been also shown to lead to the estimation of the epicentral area [R6] of an impending major earthquake. Some applications of nowcasting method have been reported in [79,84,85,86,87,88,89,90].

In the implementation proposed by [91,92], first, it is necessary to define the thresholds that will be used: The “large” earthquake, denoted as *M_λ_*, is “large” in the sense of causing damage or injuries if it occurs close. The “small” earthquakes, denoted by *M*_σ_, are counted to compute the probability of this potential. The large earthquake magnitude is selected to ensure that there are enough earthquake cycles to provide reasonable statistics. The small earthquake magnitude threshold is generally set by the completeness level of the catalog, i.e., *M_σ_ = M_c_*. The GR can be used to show that the number of small earthquakes having magnitudes larger than *M_σ_* but less than the magnitude *M_λ_* is on average a known value $\hat{N}$. The GR is then expressed as:(9)$$\hat{N}=10^{a}10^{-bM}$$
where *a* and *b* values are obtained by linear fitting in a semi-logarithmic plot for a specific seismicity region and are considered as constant along time, although this is not always the case. Denoted by $N_{c\sigma }=10^{a}10^{-bM_{\sigma }}$, the cumulative number of small earthquakes, and by $N_{c\lambda }=10^{a}10^{-bM_{\lambda }}$, the cumulative number of large earthquakes, that is, having magnitude larger than *M_λ_*, and dividing $N_{c\lambda }$ by $N_{c\sigma }$ [86], the following is obtained
(10)$$N_{c\lambda }=10^{-b\left(M_{\lambda }-M_{\sigma }\right)}N_{c\sigma }$$

The number of EQs having magnitude between *M_σ_* and *M_λ_* is used as a measure, in natural time, between two large events of magnitude *M* > *M_λ_*. The counting $N_{c\lambda }$ is the natural time, which is a prediction of the number of large earthquakes that occurs in natural time. The number of small earthquakes, occurring on an average between two large earthquakes, ${\bar{n}}_{\sigma }$, is determined. 

Setting $N_{c\lambda }=1$ in Equation (10) of [87], it is found that
(11)$${\bar{n}}_{\sigma }=10^{b\left(M_{\lambda }-M_{\sigma }\right)}$$

This means the number of small earthquakes scales exponentially with difference in magnitudes. The rates of occurrence of small earthquakes are dominated by earthquakes with magnitudes near the *M_σ_* cutoff. This rate is often inhomogeneous, as in the case of aftershock sequence, and is also discontinuous, i.e., when a main shock occurs. When we use the cumulative sum of small earthquakes, $N_{c\sigma }$, which is natural time, this is continuous, and if GR statistics are a good approximation, the natural time since the last large earthquake should be a measure of the hazard for the next $M\geq M_{\lambda }$ earthquake. Subsequently, we can obtain the earthquake potential score (EPS) for the occurrence of a large earthquake, having magnitude larger than *M_λ_*, by computing the cumulative distribution function (CDF) of small earthquakes of magnitude larger than *M_σ_* but less than *M_λ_: M_σ_* ≤ *M* ≤ *M_λ_*. The tabulation of the number of small earthquakes for each cycle of large earthquakes cycle gives us the probability density function (PDF).

## 4. Results

The results obtained from the analysis of the three sub-catalogues studied with the proposed methodologies are showed in this section. The data set were organized as follows: Part 1 corresponds to the sub catalogue from 1 January 2010, until 7 September 2017, closing with the M8.2 earthquake. Part 2 contains the seismic activity yield by the aftershocks whose elapsed time was estimated by the Utsu–Omori law, ranging from 8 September 2017 to 31 March 2018. Finally, Part 3 is the seismic activity monitored from 1 April 2018 to 31 December 2020. 

### 4.1. The Time Evolution of b-Value in GR Law

Following studies, carried out by authors such as Ma (1978) [54], to observe the variations in b-value along the time before several large earthquakes in north China, we performed the same type of analysis of b-values. Contrary to it, it is generally assumed that the b-value is considered as a constant in time for a particular region; Ma [54] observed that peaks in the absolute b-values seemed to be related to the magnitude of the impending event. However, not all temporal decreases in b-value were followed by a significant earthquake. To do this, first, we compute the b-value for catalog part 1 and part 3, while part 2 was fitted with UO law in the next section. The obtained values were for part 1: *b* = 0.985 ± 0.03 and *a* = 6.5763; for part 3: *b* = 1.3588 ± 0.02 and *a* = 8.3909. After this, we compute the GR relationship for three months intervals from January 2010 to June 2016, with a window of a month, from November 2016 to September 2017, and from March 2017 to 2021 for three months interval. 

Figure 3 clearly shows that b-values should not be considered as constants in time; moreover, the b-values fluctuates in this region from 1.57 to 2.48 along the ten years analyzed. We can observe that the b-value comes from lower to higher values as the earthquake approaches, and the peak of b-values appears in June 2017, three months before the earthquake. This is coincident with the evidence found by [93], where they report an abrupt increase in the complexity measure associated with the fluctuations of entropy under time reversal on 14 June 2017, just three months before the M8.2 earthquake. 

In 2018, Perez-Oregon et al. [68] attempted an analytical demonstration of the positive correlation between the parameters *a* and *b* of GR, which initially was proposed by [94]. These authors used seismic catalogs from 27 active seismic areas around the Earth and calculated the GR parameters *a* and *b* for all these regions. This relationship was also verified by Pérez-Oregon (2018) [94], and they go deeper inside by showing for two synthetic models that the relationship is true. They analytically deduce *a* = (4.01 − 0.02 M) *b* + log C (with M magnitude, and C constant). In our case, we verified this relationship by cross plot (Figure 4) of all data obtained along 10 years; in this case, we obtain a slope of 4.1, which coincides with previous results of those authors.

### 4.2. Utsu–Omori Law

As is well known, the identification of the correct aftershocks period is an open problem which remains unsolvable; however, a good estimation is the Utsu–Omori law (UO). Figure 5 shows the UO behavior calculated from 8 September 2017 immediately after the M8.2 earthquake. In this figure, the vertical axis is the number of earthquakes per day and the horizontal axis indicate the number of days from the aftershock activity started. The red curve fitting Equation (2) obtained a good fitting with *p* = 0.6982, k = 716.1574 and c = 3.1; additional, the spent time is 210 days, and thus, under this criterion, the aftershock period extends as long as six months, with the dates selected above for part 2.

After 160 days, the hyperbolic decay attained around 15 EQs per day, that can be considered the background seismic activity in the region. Moreover, in the cumulative distribution plot of Figure 1 it is also observed a linear rate for the number of quakes after that period.

### 4.3. Multifractal Analysis

The MFDFA was applied by considering two ways: in the first one, the calculation was performed year by year by considering the whole period analyzed, and in the second, the three sub-catalogues, part 1, part 2 and part 3, were analyzed independently, one by one, taking into account their respective magnitude completeness *M_c_*. The first stage is based on the analysis of q-order fluctuation function *F_q_*(*s*). The better polynomial detrending degree was determined as *m* = 2 for all cases. The generalized Hurst exponent, *H*(*q*), is a measure of the multifractality, and if *H*(*q*) is constant for all *q*-values, the process is monofractal. In Figure 6a, the behavior of *H*(*q*) vs. *q* is showed for the three sub-catalogues, parts 1 (blue), 2 (red) and 3 (yellow), where the multifractality is revealed. Part 2 corresponds with the aftershock activity. In comparison, *H*(*q*) shows values greater for part 1 than part 3 for *q* < 0; however, for *q* > 0 values, both parts behave in a similar way. By considering the yearly *H*(*q*) behavior, in Figure 6b the differences Δ*H = H*(−5) − *H*(5) are plotted. In this plot, it can be observed that multifractality decreases; this decrease is most pronounced after the M8.2, where such differences are less than 0.03, which suggests that the behavior becomes monofractal. The *H*-values (*H* < 0.5) shows that antipersistency is present in the seismicity analyzed for the three cases.

On the other side, the multifractal or singularity distribution *f*(*α*) is obtained by a Legendre transform. Some parameters to characterize the multifractality are the width Δ*α*, *α*_0_ where *f*(*α*_0_) is maximum and the *f*(*α*) symmetry of the multifractal spectra. The fractal distribution was calculated for the three parts which are showed in Figure 7a, where the three spectra are compared. It can be observed that *α_0_* coincides for parts 1 and 3, but for part 2 this value is a bit larger; nevertheless, in the three cases, their respective values do not indicate random processes (0.1 < *α*_0_ < 0.22). The width Δ*α* was analyzed for each year, as is showed in Figure 7b, where it is observed a decrease from 2011. From 2017 to 2018, an absolute minimum was attained in the aftershock period; this result agrees very well with Telesca and Lappenna (2006) [7], who analyzed the seismic activity of central Italy and found a sudden decrease in multifractality just after the strong earthquake that struck that area in 1997, as is showed Figure 6 of [7], that is, the multifractality decreased, but it recovered from the end of 2018 until the end.

Regarding the symmetry, only part 1 slightly displays an asymmetry to the left.

### 4.4. Visibility Graph Analysis

The connectivity as magnitude function behaves as a linear relationship, similar to that reported by [61]; in Figure 8a, it is observed that the connectivity increases. The plot in Figure 8b shows the connectivity, that depends on the dynamical features of Tehuantepec seismicity, showing a sustained increase per year, except for 2017, when the M8.2 stroke arrives.

Applying the VG method, the degree k was calculated for part 1, part 2 and part 3, considering all events (blue line in Figure 8b) and with magnitude ≥ *M_c_* (red line in Figure 8b). For any event of each magnitude sequence, the degree k was calculated as the number of links between that event with any other events of the sequence, on the basis of the rule defined in Equation (8). We analyze the k–*M* plots, that is, the relationship between the magnitude of each event and its degree k. The relationship between the magnitude *M* and the degree k is obtained fitting the slope of the right line of k–*M* relationship, by using a least square method (Figure 8a), this procedure was performed for each year, and the slope values calculated yearly are plotted in Figure 8b.

### 4.5. Nowcasting Analysis

We apply the nowcasting method to the seismic data series of two sub-catalogues, parts 1 and 3. We took as the smallest magnitude, *M_σ_* = 3.7, corresponding to completeness catalog for part 1 and to 3.8 for part 3. The large magnitude was selected in order to have enough amount of EQs to compute the statistical distributions between two large EQs; in the case of part 1 we choose *M_λ_* ≥ 4.9, and then we have 91 earthquake cycles; for part 2, we select *M_λ_* ≥ 4.7.

The computed GR for the total catalog part 1 is b = 0.985 ± 0.03, which is close to 1. While the computed value for part 3 is b = 1.3588 ± 0.03, as we can see, the *b*-value is not constant in time, as was illustrated in Section 4.1, even when it comes from the same region. These values denote important differences between before and after the M8.2 earthquake.

Following the methodology proposed by Luginbuhl et al. 2018 [69], we plot the cumulative number of small earthquakes, *N_cλ_*, with magnitudes *M_σ_* ≥ 3.7 and 3.8 (part 1 and part 3, respectively) versus the cumulative number of earthquakes, *N_cσ_*, with *M_λ_* ≥ 4.7; in natural time (*N_cλ_* vs. *N_cσ_*), the best fit least squares the slope to the straight line passing through the origin, which is used to compute the nowcasting. For part 1 and part 3, the linear fit displays a slope of 0.0059 and 0.0047, respectively. Subsequently, we performed the same plot against ***t*** clock time in days; if the rates of seismicity were constant, this would be well approximated by *dN_cσ_/dt*. 

Finally, we obtain the EPS for large earthquakes with magnitudes *M_λ_* ≥ 4.7 (4.9), versus the number n_s_ of small earthquakes of magnitude *M_σ_* such that (3.7 or 3.8 ≤ *M_σ_* < 4.7) between two large earthquakes for the two sub-periods considered. In order to compare the behavior between the nowcasting with a Poisson distribution, since these have the same meaning, the last is showed in the plot with blue points. 

For part 1, in the graph (Figure 9), we can see when more than a hundred small earthquakes have occurred, an EPS > 70% is achieved. Moreover, before the M8.2 earthquake on 7 September 2017, one can count that (ns = 250) earthquakes have taken place leading to an EPS that increases suddenly from 70 to 97% in favor of the last strong EQ; previously in the graph, a second step is observed where the probability increases arriving until 87% around 175 events. Figure 9a shows the histogram with a maximum frequency of 11 EQ cycles.

For part 3, the graph (Figure 10) shows great coincidences between Poisson and CDF curves until 300 small earthquakes, and when (ns = 400) earthquakes have happened, the EPS of a strong EQ increases, from around 70% to 88% for an increase of a hundred small earthquakes. Moreover, before the M7.4 earthquake on 23 June 2020, the EPS arrived at 88%. In this case, the linear dependence between the cumulative number of small earthquakes versus the cumulative number of earthquakes *M_λ_* ≥ 4.7 was assured (rms > 96%), which indicates the seismicity rate remained constant, which was not the case in part 1 (rms ≈ 80%).

## 5. Discussion

The zone of the Gulf and Isthmus of Tehuantepec deserves special attention, mainly after the M8.2 earthquake in 2017, which has been followed by hundreds of aftershocks with a clustered spatial distribution, where small earthquakes keep happening. Recently, Richards (2020) [95] has showed that in some parts of the world, the trench-parallel distribution of aftershocks appear to be spatially constrained by fractures on the subducting oceanic crust. The case in concern precisely shows these characteristics, where the spatial distribution models of subduction zone aftershocks can be used to better forecast the lateral (along-arc) extent of damaging aftershock swarms following large magnitude subduction-related earthquakes. 

An important finding was that the Gutenberg–Richter law indicates the b-values are not constant along time in the same studied area and that the completeness magnitude values changes over the time. We obtain the b-values which fluctuate in this region from 0.57 to 2.48 along the ten years analyzed, which overlap with the b-values published by [59] from 0.5 to 1.5. According to [45], the b-values that exceed 1 are often found in areas with increased geological complexity, which is a possible indicator of multi-fracture areas, areas which have also experienced slip, which is the case of the studied region.

On the other hand, [95] showed that a low b-value is closely related to the low degree of heterogeneity of the cracked medium, enormous stress and strain, high deformation rates, large faults, and thus, seismic moment rates suggesting as possible regions that are subjected to higher applied shear stress after the mainshock. Concerning the relationship between the a and b parameters of GR, we verified a positive correlation between both parameters, which was initially proposed by Bayrak et al. (2002) [96] and corroborated by Perez-Oregon et al. (2018) [68,94], the slope fitted by cross plot of all data obtained along 10 years, was 4.1. These authors also highlight the relationship between the number of earthquakes greater than M and related to rupture area larger than S per year, which was reported by Kanamori and Anderson (1975) [97], where M and S are related as log S = M − 4. Subsequently, Aki [98] suggested that the fractal dimension of regional or worldwide seismic activity is simply twice the b value; however, it is not constant. While the linear relationship between b and a is linear with slope 4 and following the arguments of Legrand (2002) [99], twice the b value is only valid for the case of earthquakes of intermediate magnitude, but for small events, the appropriated relation must be D = 3*b*, and for large events D = *b*, so we can assume this last relation for the fractal dimension of the Tehuantepec region.

Regarding the multifractal analysis, the generalized Hurst exponent, and the singularity distribution *f*(*α*), were calculated for *q* = −5 to 5. The first calculation was performed for the three sub catalogues (part 1, part 2 and part 3), with the aim to compare dynamical features before the M8.2 (part 1 is the left segment in Figure 1), the aftershocks period (part 2) and finally the behavior during the regular seismicity activity (part 3). From Figure 6a, *H*(*q*) displays a nonconstant behavior for the three parts, which means the multifractality is present, which is confirmed with the singularity spectra showed in Figure 7a. Two important issues must be noted: the seismic activity displays an antipersistent behavior whilst the width Δ*α* changes, being the shortest in part 2, the aftershocks period. In order to identify the changes in the multifractality, the widths Δ*H* and Δ*α* were estimated yearly along the whole period analyzed. Figure 6b and Figure 7b show Δ*H = H*(−5) − *H*(5) and Δ*α* for each year. The first value (for 2010) is low because the number of earthquakes is around 700 events; however, from the year 2011, Δ*H* and Δ*α* increased. The subsequent values for the following years decreased, suggesting that multifractality is being lost, in particular after the aftershocks period; nevertheless, Δ*α* remained low but constant.

In a previous work [12], the multifractal properties of the earthquake magnitude series of seismicity occurring in the period 2005–2010 on the Mexican South Pacific Coast were investigated for five regions defined along the Pacific Mexican coast where the Tehuantepec Gulf was considered. In that investigation, it was found that Δ*Hq* = 0.2078 for the Tehuantepec Isthmus and showing antipersistence; nevertheless, in a comparative behavior, for the Kachchh, western India [13], a similar analysis conducted for the period 2003–2012 showed persistence and Δ*Hq* ≈ 0.11. 

The connectivity, which is a measure of the degree distribution of the nodes, was determined by the visibility graph method, and was performed yearly. The study is based in the k-degree (connectivity) vs. magnitude plane. Figure 8 shows only the behavior observed in 2017 displaying their connectivity vs. magnitude. Here, it can be observed that the node associated with the M8.2 (red point located in the right superior angle) has the maximum connectivity. A measure of this relationship is the slope of the linear fitting, which indicates the correlation between both parameters. In Figure 8b, the slope k-fit observed each year was displayed. In this figure there are two curves: the blue curve is the connectivity estimated with all seisms without considering the completeness magnitude, whilst the red curve was calculated by taking into account the earthquakes with *M*
≥
*Mc.* In both cases, the connectivity increases as a linear relationship, but with a decreasing trend in 2017. In comparison with the multifractality, while it decreases, the connectivity increases, but in both cases the presence of the mainshock changes that behavior, decreasing both.

The nowcasting and multifractal analysis allows us to identify differences between the dynamical features associated with temporal changes, but we can also observe the signature of the aftershock swarms following the spatially constrain, due to the exposed Tehuantepec Transform/Ridge that collides with the Middle America Trench off Chiapas and continues to the north within the continental plate.

From the M8.2 earthquake, which struck on 7 September 2017, the seismic activity changed, as is observed in Figure 1 and Figure 2: the temporal rate cumulative occurrence of earthquakes increased suddenly, and the spatial distribution of earthquakes changed to a clustered linear distribution, after the M8.2. This distribution can be associated with the collision of the exposed Tehuantepec Transform/Ridge with the Middle America Trench off Chiapas and follows in the continental plate (See Figure 9 in [37]). According to [19], the deformation may perhaps be related to the variation in the geometry of the Tehuantepec Transform/Ridge, in that when it was being subducted, it induced deformation in the upper part of the North America plate, which is the intersection between Tehuantepec Transform/Ridge and Trench.

The nowcasting analyses revelated the EPS: For part 1, before the M8.2 earthquake on 7 September 2017, one can count that (ns = 250) EQs have taken place, leading to an EPS that increases suddenly from 70 to 88% in favor of the last strong EQ. The corresponding histogram shows a maximum frequency of 11 EQ cycles. For part 3, before the M7.4 earthquake on 23 June 2020, it also shows two steps, and the EPS indicates the number of earthquakes required to reach approximately eighty percent probability needs to be greater than 400, and it behaves similarly to the Poisson distribution before 300 EQs.

## 6. Conclusions

In light of our goal, we studied the sequence of magnitudes of the earthquakes occurred within the Isthmus of Tehuantepec in southern Mexico, from 1 January 2010 to 30 July 2020, by using GR relationship, UO law, multifractal detrended fluctuation analysis, visibility graph and the nowcasting method.

We obtain that the Gutenberg–Richter law indicates three b-values and two completeness magnitude values: the first values, part 1: *b* = 0.985 ± 0.03, *a* = 6.5763 and M_c_ = 3.7; for part 3: *b* = 1.3588 ± 0.02 and *a* = 8.3909 and M_c_ = 3.8. We verified a positive correlation between the *a* and *b* parameters of GR: the slope fitted by cross plot of all data obtained along 10 years was 4.1; as the *b*-value is not constant, the *b*-values fluctuate in this region from 0.57 to 2.48. The *b*-value fluctuates from lower to higher values as the M8.2 earthquake approaches, and the peak of the b-values appears in June 2017, three months before the earthquake.

Concerning the multifractal characteristics between the three sub-catalogues, we point out the following findings:

The generalized Hurst exponent, *H*(*q*), of the three sub-catalogues indicates antipersistent behavior, because 0.1 < *H*(*q*) < 0.25. In comparison with previous studies, where persistence activity was observed, in our case we found antipersistence, which indicates that short fluctuations are dominant. In part 2, it was showed that the shorter range of *H*(*q*) suggesting a loss of multifractality, associated with the aftershocks.

The magnitude sequence of parts 1 and 3 are more multifractal than that of the sub-catalogue part 2. This indicates that the magnitude sequence before the M8.2 and six months after, are more heterogeneous, suggesting a likely instability in the seismic activity, possibly associated with the tectonic activity developed before and after the M8.2 main shock.

The connectivity k increases as a linear relationship but with a decreasing in the 2017 year.

The nowcasting analyses revelated the earthquake potential score, EPS, increases suddenly from 70 to 88% in favor of the last strong EQ before the M8.2 earthquake on 7 September 2017, when one can count that (ns = 250) EQs have taken place.

The studies carried out, based on nowcasting method and in the multifractality, confirm the observations in the data that show changes in underlying dynamics in the intersection between Tehuantepec Transform/Ridge and the Middle America Trench.

## Figures and Tables

**Figure 1 entropy-24-00480-f001:**
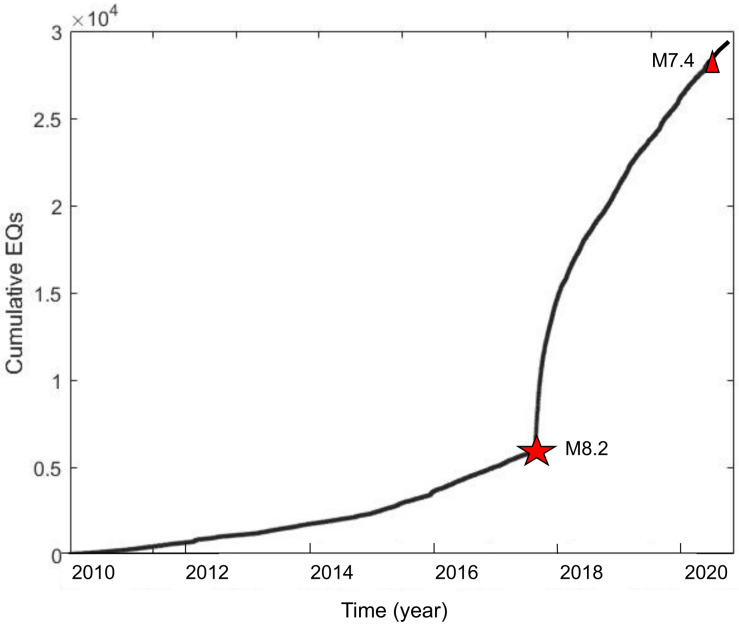
Cumulative earthquakes distribution along a ten-year period from 1 January 2010 to 31 December 2020; the two large events are showed: M8.2 on 7 September 2017 (red star), and M7.2 on 23 June 2020 (red triangle).

**Figure 2 entropy-24-00480-f002:**
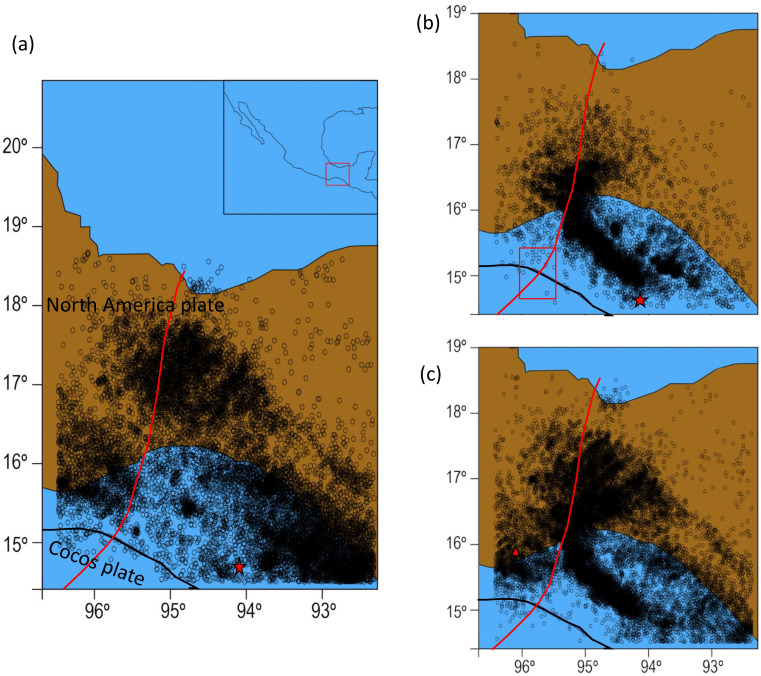
Spatial distribution of earthquakes along the Gulf and Isthmus of Tehuantepec: (**a**) part 1, (**b**) part 2 and (**c**) part 3. Red star and triangle represent the epicenters of the M8.2 and the M7.4 earthquakes, respectively. Black line depicts the Middle America Trench. Red line indicates the inferred location of the Tehuantepec Transform/Ridge, and the red rectangle indicates collision of the Tehuantepec Ridge with the Middle America Trench.

**Figure 3 entropy-24-00480-f003:**
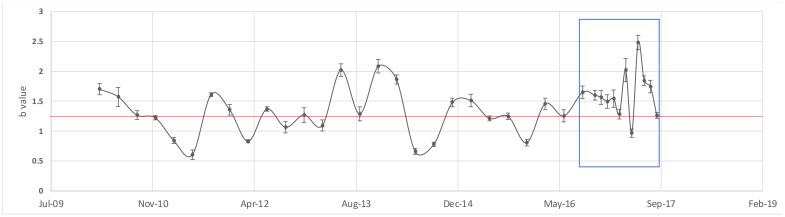
Evolution of b-value obtained by GR law for by three months interval from 2010 until August 2016, and from January to September 2017. The b-values per month are indicated by a blue box.

**Figure 4 entropy-24-00480-f004:**
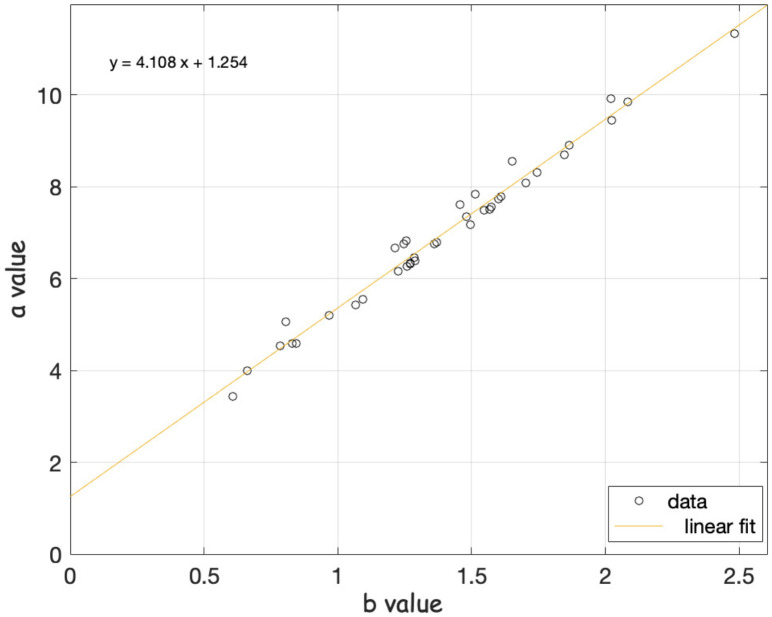
Scatter plot of b-value and y-intercept *a* obtained by GR law for all periods analyzed.

**Figure 5 entropy-24-00480-f005:**
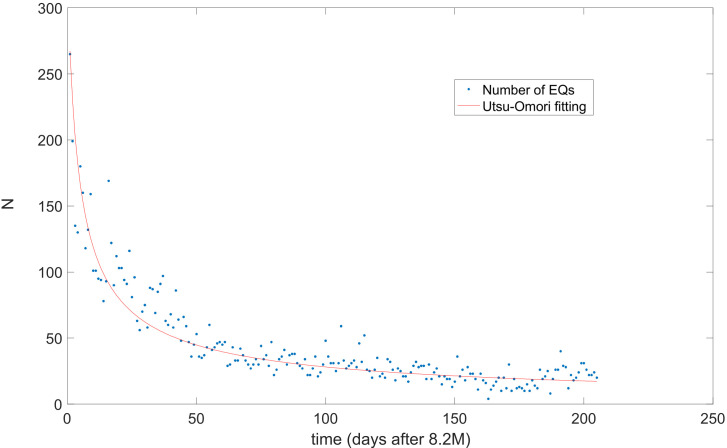
Fitting of Utsu–Omori law of Equation (3), where *p* = 0.6982, k = 716.1574 and c = 3.1.

**Figure 6 entropy-24-00480-f006:**
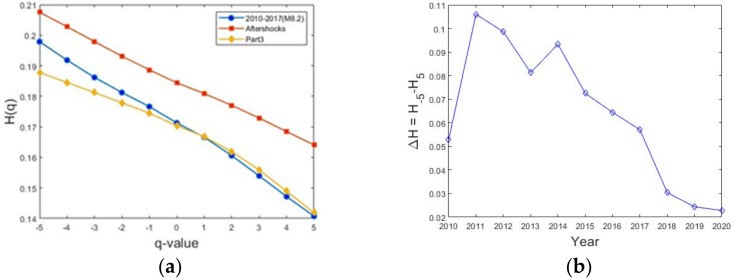
Generalized Hurst exponent calculated for *q* = −5 to 5. (**a**) The red curve corresponds with the aftershocks period. (**b**) Temporal sequence of Δ*H* = *H*_−5_ − *H*_5_ which suggest that multifractality is decreasing, in particular after the aftershocks period.

**Figure 7 entropy-24-00480-f007:**
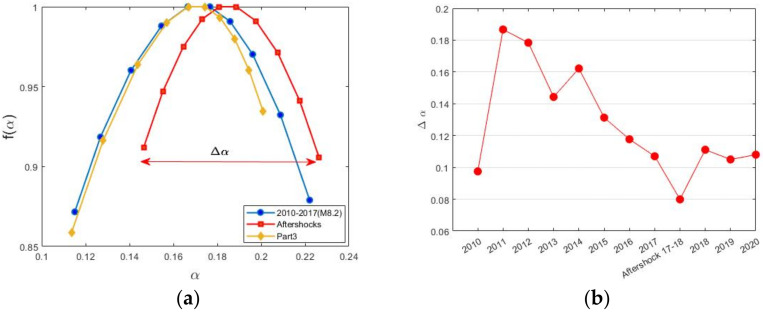
(**a**) Singularity spectra for the three parts. (**b**) Δ*α* behavior calculated versus each year is cumulated. The minimum value of Δ*α* occurred during the aftershocks period.

**Figure 8 entropy-24-00480-f008:**
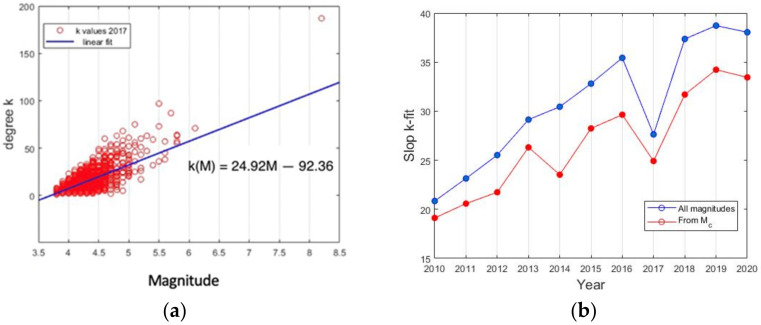
(**a**) Connectivity vs. magnitude; this plot is an example of the general behavior between both parameters. The slop of the fit represents a linear relationship; (**b**) The slop of k-degree per year shows a linear increasing; nevertheless, in 2017 this value falls.

**Figure 9 entropy-24-00480-f009:**
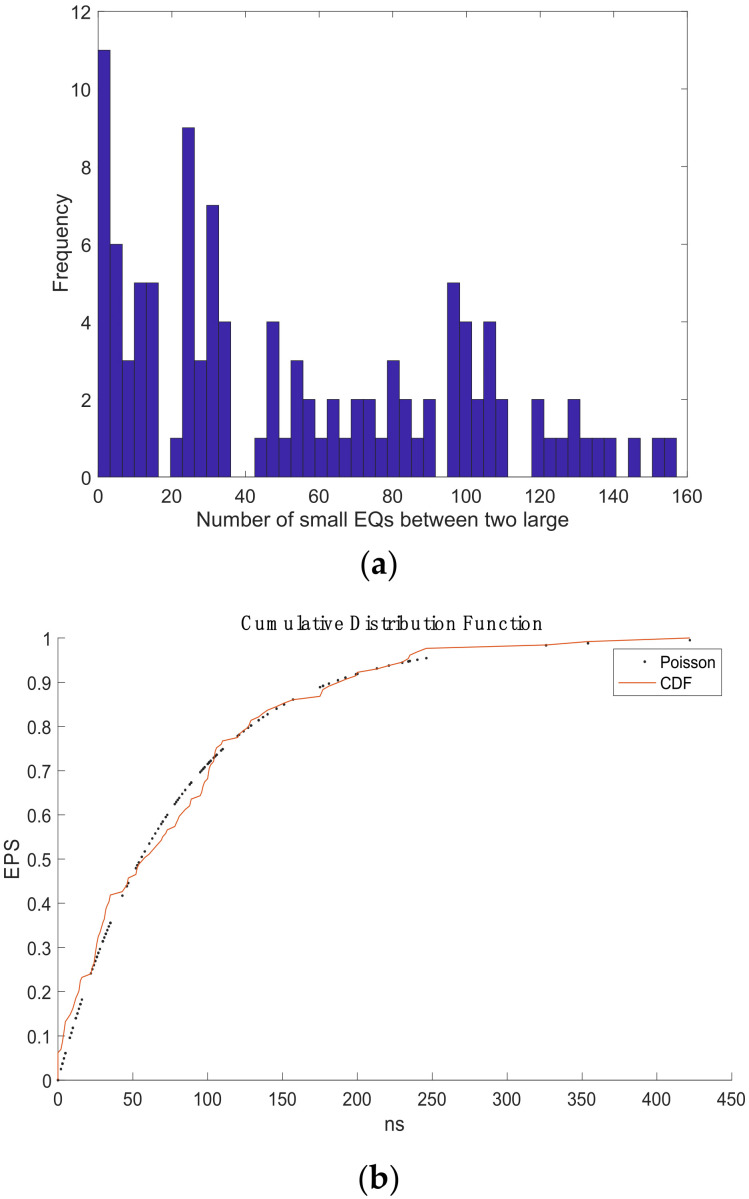
(**a**) the histogram of part 1; (**b**) The earthquake potential score (EPS) obtained from nowcasting method.

**Figure 10 entropy-24-00480-f010:**
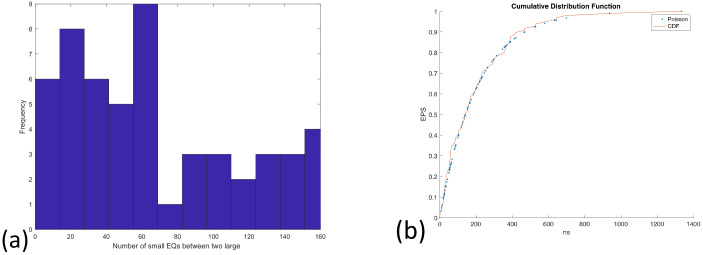
(**a**) the histogram of part 3; (**b**) The earthquake potential score (EPS) obtained from nowcasting method follows Poisson behavior.

## Data Availability

Not applicable.

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
