# Peer review of "Nonlinear Statistical Features of the Seismicity in the Subduction Zone of Tehuantepec Isthmus, Southern México"

_entropy, 2022, doi:10.3390/e24040480_

Round 1

Reviewer 1 Report

1) Line 82: change "reason" with "period"

2) line 103: change "completes" with "completeness"

3) line 392: set the b-value as positive, since this is the standard way to represent it

4) lines 393-395: it is not clear "every month": do you mean "with a shift of 1 month"?

5) line 396: Fig. 3 only shows the b-value and not a-value; please rephrase accordingly

6) Fig. 3: it would be helpful to add the error bar for each calculated b-value

7) line 440: set the p-value as positive

8) lines 485-487: the authors find a strong decrease of multifractality within the aftershock period. Thsi results agrees very well with Telesca and Lapenna (Tectonophysics, 423, 115-123), who analysed the seismic activity of central Italy and found a sudden decrease of multifractality just after the strong earthquake that struck that area in 1997 (Fig. 6, Tectonophysics, 2006). the authors may cite this paper commenting on their results.

9) line 513,514: set b-value as positive

10) The Discussion seems to repeat the same comments made on the presented results. Thus the authors can choose to keep the comments in the Results section and delete the Discussio section or to keep the Discussion section and delete identical sentences or comments written in the Results section.

11) Conclusions: set b-values as positive

Author Response

Thank you very much for your comments and suggestions about our manuscript entitled “Nonlinear statistical features of the seismicity in the subduction zone of Tehuantepec Isthmus, southern México” (No. Entropy-1639176). We have reviewed each of the comments very carefully and made corrections to the manuscript. which, we believe, has been improved and we hope that it will be approved for publication. It is important to mention that some of your comments coincide with comments made by other reviewers, so we take all suggestions into account. We send the revised manuscript with colors showing the corrections and  the manuscript without colors with all the changes and corrections incorporated.

Reviewer 2 Report

In this paper the authors analyze the earthquake time series in the subduction zone of Tehuantepec Isthmus (southern Mexico) during 2010-2020, by means of the Gutenberg-Richter scaling relation, Omori-Utsu relation, multifractal detrended fluctuation analysis, visibility graph and nowcasting. In the title, the authors mention nonlinear features of seismicity, but I am not sure if they justify it in the manuscript. The paper presents some interesting results; however, it needs extensive editing in its structure, graphs and language. Therefore, I would recommend major revisions if the authors are willing to undertake the work required and address the following several comments. In the attached manuscript, there are also several highlights of the text that needs correction and/or rephrasing. 

1) In the Introduction section, the list of references is limited to particular authors-groups (e.g., lines 42-45). Provide a more complete list of references.

2) Lines 48-50 and 52-54, provide reference(s).

3) Lines 55-56. Explain why the SOC theory implies that earthquakes are unpredictable and provide appropriate references.

4) Lines 57-58. Provide references and discuss which are the possible statistical methods that can identify precursory signals.

5) Lines 70-71. The authors say that the spatial distribution of seismicity before the M8.2 earthquake was homogeneous. Describe in which way.

6) Fig.1 shows the cumulative number of earthquakes with time. The cumulative number increases significantly after the M8.2 earthquake, as expected from the large number of after hocks, but then the rate remains high. Is there a possibility that this reflects the increased detectability of the seismic network? Discuss further this issue and also add in the figure the cumulative number of events with magnitudes above the magnitude of completeness.

7) Add in Fig.2 the epicenters of the M8.2 and the M7.4 earthquakes. Seismicity should be scaled according to magnitude. In the text, the authors name various features that are not displayed in Fig.2 e.g. lines 125-131.

8) Line 83. What do the authors mean by unusual behavior in this zone?

9) Line 90. The authors say that the seismicity dynamics during the studied period may indicate increased seismic risk. In which way, since two large earthquakes already struck the region.

10) Lines 96-98. Provide references.

11) Line 98. The authors mention nowcasting without introducing the method first. The same for the other methods that are used (lines 100-102). Introduce the methods first, explain why you chose those methods and what is expected from the results.

12) Lines 110-113. The authors refer to multifractal analysis performed on the region during 2005-2010 that is outside the studied period (2010-2020). Do they refer to some previous study?

13) Lines 114 and 116. What is Hq? Introduce it first.

14) Lines 103-119. The whole paragraph discusses the results rather than introducing the methodology. The results should be discussed after the analysis. 

15) Line 163. Provide the full name of UNAM first.

16) Section 3.2. Provide the mathematical relationship of the Gutenberg-Richter law.

17) In Section 3 the methodology is introduced. In these sections the authors also discuss the results of previous studies in respect to their study, a discussion that should be placed either in the introduction or in the Discussion section. Revise accordingly.

18) Line 226. The authors use throughout the text the terms “seisms”, “quakes”, “earthquakes” etc. It is better to use only one word for consistency.  

19) Lines 228-230 are repetition of lines 224-226.

20) Lines 237-238 “which fluctuate enclosed by earthquake sequences”. What do the authors mean? Rephrase.

21) Line 245. Provide reference(s) for the range of p-values.

22) Lines 255-258. Provide references. In the past years there has been much work accomplished on the multifractal structure of earthquake time series. Provide a more representative list of references on this matter.

23) Line 347. The mathematical relation is missing. The same in line 353.

24) Line 349. The a and b values are usually estimated with the maximum likelihood method and not by linear fitting. In addition, the plot is semi-logarithmic.

25) Line 354-356. Rephrase the whole sentence and define the occurrence of an event in “natural time” domain.   

26) Line 359. Eq.9 is missing.

27) Why the authors did not calculate the a and b values for Part 2 (the aftershock sequence)? Provide the frequency-magnitude plots for Part 1, 2 and 3 and the fitting according to the GR law. Also discuss the magnitude of completeness for each part and how it was calculated. In lines 392-393 provide the confidence intervals for the a-values and keep the numbers up to two decimal digits, according to the confidence intervals.

28) Lines 393-395. The time segments are not consistent.

29) The GR a-values are not shown in Fig.3, as the authors claim in line 396. In Fig.3 give the axis names and also show the confidence intervals of the estimated b-values.

30) The value of 2.48 for b is quite high. Such high values are usually met in volcanic regions or earthquake swarms. Can the authors explain if this value is supported by the data? In the red window of Fig.3, the b-value shows significant fluctuations, and the authors should show that these fluctuations are “real” and not a malfunction of the calculations. The peak rather seems to be in August 2017 rather than in June 2017 (line 399).

31) Lines 421-425. Rephrase, that sentence doesn’t make sense. Actually, the whole paragraph (lines 409-425) should be moved to the Discussion section.

32) In Fig.4 give the name of the axis and show the linear regression relationship between a and b.

33) Lines 435-436 “…open problem unsolvable”. Rephrase and provide references.

34) line 437. 2017 not 2027.

35) Line 440. Provide confidence intervals for the estimated values and make sure you are consistent with decimal digits.

36) Line 441. Which criterium? The authors show in Fig.5 the number of aftershocks for 210 days and then they say that the aftershock period extents to six months.

37) In section 4.3, MFDFA is applied to the interevent time series?

38) Give the confidence intervals of H(q) in Fig.8a. The H(q) values are close so that multifractality may not hold.

39) In Fig.8b, is the analysis performed in cumulative time windows? The 2019-2020 values are quite low.

40) In Fig.8a the results show antipersistent behavior, in contrast to other studies that show H(q) values greater than 0.5. Discuss the results in comparison to previous studies in the Discussion section. It would also be useful to add a graph showing how the fluctuation function scales with the time segments to show the low slopes (H(q)).

41) In section 4.4 explain in more detail how the results were obtained.

42) Lines 511-512. How did the authors choose the Mλ values? Why didn’t they choose a larger value, for instance?

43) Line 523. Show how the slope values were obtained.

44) Line 545. Is there Fig.11? What do the authors mean by “particular behavior”?

46) Lines 566-567. The authors say that the magnitude of completeness changes over time, but this is not shown. The authors just mention different values for the different parts without justification. The authors should explain how they estimate the magnitude of completeness and calculate it in the same time windows as the analysis of the b-values.

46) Repetition in Lines 579-587. These lines are exactly the same as lines 417-425.

47) Previous studies have also demonstrated the loss of multifractality during aftershock sequences. Refer to these studies and compare your results.

48) Line 606. Figure 1? Probably wrong reference.

49) Lines 652-653. Shorter range of H(q) values rather indicate monofractal structure rather than randomness. In addition, aftershocks are clustered sequences showing non-random behavior. Revise accordingly. 

Author Response

(The authors gave the same response as above.)

Reviewer 3 Report

Please, see my report attached.

Author Response

(The authors gave the same response as above.)
